# Nerfin-1 represses transcriptional output of Hippo signaling in cell competition

**Pengfei Guo, Chang-Hyun Lee, Huiyan Lei, Yonggang Zheng, Katiuska Daniela Pulgar Prieto, Duojia Pan\***

Department of Physiology, Howard Hughes Medical Institute, University of Texas Southwestern Medical Center, Dallas, United States

**Abstract** The Hippo tumor suppressor pathway regulates tissue growth in *Drosophila* by restricting the activity of the transcriptional coactivator Yorkie (Yki), which normally complexes with the TEF/TEAD family DNA-binding transcription factor Scalloped (Sd) to drive the expression of growth-promoting genes. Given its pivotal role as a central hub in mediating the transcriptional output of Hippo signaling, there is great interest in understanding the molecular regulation of the Sd-Yki complex. In this study, we identify Nerfin-1 as a transcriptional repressor that antagonizes the activity of the Sd-Yki complex by binding to the TEA DNA-binding domain of Sd. Consistent with its biochemical function, ectopic expression of Nerfin-1 results in tissue undergrowth in an Sd-dependent manner. Conversely, loss of Nerfin-1 enhances the ability of winner cells to eliminate loser cells in multiple scenarios of cell competition. We further show that INSM1, the mammalian ortholog of Nerfin-1, plays a conserved role in repressing the activity of the TEAD-YAP complex. These findings reveal a novel regulatory mode converging on the transcriptional output of the Hippo pathway that may be exploited for modulating the YAP oncoprotein in cancer and regenerative medicine.

DOI: https://doi.org/10.7554/eLife.38843.001

## Introduction

The Hippo signaling pathway is a conserved mechanism that regulates organ size, tissue regeneration and stem cell biology in diverse animals (*Halder and Johnson, 2011*; *Harvey and Tapon, 2007*; *Pan, 2010*; *Zhao et al., 2010*). Central to this pathway is a kinase cascade comprising tumor suppressors Hippo (Hpo, MST1/2 in mammals) and Warts (Wts, LATS1/2 in mammals), which are activated at the cell cortex by various upstream inputs. Wts/LATS in turn phosphorylates and inactivates the oncoprotein Yorkie (Yki, YAP/TAZ in mammals). As a transcriptional coactivator, Yki/YAP/TAZ does not bind DNA directly and its ability to regulate target gene expression relies on its obligatory DNA-binding partner encoded by the TEF/TEAD family transcription factor Scalloped (Sd, TEAD1/2/3/4 in mammals). Consistent with the importance of the Sd/TEAD-Yki/YAP/TAZ transcription factor complex in Hippo signaling, loss of Sd completely rescues Yki-induced overgrowth, and mutations in Yki/YAP/TAZ that disrupt its physical interactions with Sd/TEAD abolishes the growth-promoting activity of Yki/YAP/TAZ.

Given its critical role in dictating transcriptional output of the Hippo pathway, there is great interest in understanding the function and regulation of the Sd/TEAD-Yki/YAP/TAZ complex. Recent studies in *Drosophila* have led to a default repression model concerning Sd function: in the absence of Yki, Sd functions by default as a transcriptional repressor that actively represses the transcription of Hippo target genes, and Yki promotes growth by de-repressing Sd's repressor function (*Koontz et al., 2013*). This model provides a plausible explanation for the perplexing observation that while Yki is required for normal tissue growth, loss of Sd has a negligible effect in growth in most *Drosophila* tissues: unlike loss of Yki, which leads to repression of Hippo target genes and

\*For correspondence: Duojia.Pan@UTSouthwestern.edu

**eLife digest** Animals uses a range of mechanisms to stop their organs from growing once they have reached the right shape and size. One of these processes, a set of chemical messages called the Hippo pathway, controls the balance of cell death and cell division. In fruit flies, Hippo works by repressing a complex formed of two proteins, Yorkie and Scalloped, which normally switch genes on to encourage cells to grow. Yorkie is also involved in cell competition, a process in which cells in a tissue compare themselves to each other. Healthier 'winner' cells then kill neighboring 'loser' cells that are weaker or damaged. This ensures that the tissue keeps working properly.

Despite Yorkie and Scalloped being key to control the growth and health of tissues, how the activity of these proteins is regulated was not well understood. To investigate, Guo et al. conducted a series experiments on fruit flies and found that a protein called Nerfin-1 can bind onto Scalloped to stop the Scalloped-Yorkie complex from switching on genes. As a result, flies with too much Nerfin-1 had stunted tissue growth. In addition, Guo et al. confirmed that the Nerfin-1 equivalent in mammals acts in the same way. Further work revealed that Nerfin-1 also plays a role in cell competition: without this protein, 'winner' cells became 'super winners', eliminating even more of the loser cells.

Besides regulating the size of organs, the Hippo pathway is also involved in stopping cells from dividing uncontrollably and becoming cancerous. Further research may therefore focus on Nerfin-1 and its equivalent in mammals to understand how this protein could contribute to the emergence of cancer.

DOI: https://doi.org/10.7554/eLife.38843.002

tissue undergrowth, loss of Sd would lead to de-repression of Hippo target genes and therefore a much weaker effect on tissue growth. Indeed, despite its negligible effect on normal tissue growth, loss of *sd* completely rescues the undergrowth phenotype caused by loss of *yki* (*Koontz et al., 2013*). Further support for this model came from the identification of an Sd-binding protein called Tondu-domain-containing Growth Inhibitor (Tgi, Vgll4 in mammals) (*Koontz et al., 2013*), which competes with Yki to bind to the C-terminal region of Sd in a mutually exclusive manner. As expected of a Sd corepressor, loss of *tgi* rescues the undergrowth phenotype of *yki* mutant cells. However, unlike the full rescue of *yki* mutant by loss of *sd*, the rescue by *tgi* is partial, suggesting the existence of additional co-repressor(s) of Sd (*Koontz et al., 2013*). Identification of such corepressors should provide important insights into transcriptional control of the Hippo signaling pathway.

Cell competition was first described in *Drosophila* (*Morata and Ripoll, 1975*) whereby underperforming cells (aka loser cells), such as those with reduced ribosomal activities (the *Minute* mutations), are actively eliminated by cell death when juxtaposed with wildtype cells (aka winner cells) (*Moreno et al., 2002*). It has since been extended to many additional contexts involving social interactions between cells of different fitness, such as the elimination of neoplastic tumor cells by neighboring wildtype cells, the elimination of cells lacking the Dpp receptor TKV by their wildtype neighbors, or the elimination of wildtype cells by cells with higher Myc activity (*de la Cova et al., 2004*; *Moreno and Basler, 2004*; *Moreno et al., 2002*; *Rhiner et al., 2010*; *Yamamoto et al., 2017*). Recent studies further suggested that cell competition is conserved in mammals and may contribute to diverse physiological processes such as embryogenesis and tumor suppression (*Gogna et al., 2015*). Several lines of evidence have implicated the Hippo signaling pathway in cell competition. It was reported that cells with higher Yki, like those with higher Myc, can eliminate their wildtype neighbors (*Neto-Silva et al., 2010*; *Ziosi et al., 2010*). Furthermore, increased Yki activity could rescue the elimination of neoplastic tumor cells or *Minute* cells by their wildtype neighbors (*Chen et al., 2012*; *Menéndez et al., 2010*; *Tyler et al., 2007*). Lastly, the TEAD transcription factors were implicated in Myc-mediated cell competition in cultured mammalian cells (*Mamada et al., 2015*). A caveat of these studies is that they often involve conditions in which Yki is massively activated at supraphysiological level. Whether Yki is required for cell competition at its endogenous physiological level remains an open question.

Here, we describe the identification of Nerfin-1 as a transcriptional repressor that antagonizes the Sd-Yki complex by binding to the TEA DNA-binding domain of Sd. Not only does ectopic expression

of Nerfin-1 result in tissue undergrowth in an Sd-dependent manner, loss of Nerfin-1 enhances the ability of winner cells to eliminate loser cells in multiple scenarios of cell competition. We also provide evidence showing the conserved function of a mammalian ortholog of Nerfin-1 in repressing the activity of the TEAD-YAP complex.

## Results

### Nerfin-1 binds to Sd and antagonizes transcriptional activity of the Sd-Yki complex

In an effort to identify additional regulators of the Sd-Yki transcription factor complex, we searched *Drosophila* Interaction Database (DroID, http://www.droidb.org) for proteins that physically associate with Sd and/or Yki. Nerfin-1 emerged as a candidate that interacts with both Sd and Yki. Specifically, a genome-wide yeast-two-hybrid screen using Nerfin-1 as bait identified Sd as a Nerfin-1-binding protein (*Giot et al., 2003*). In addition, a large-scale affinity purification screen in *Drosophila* S2R+ cells identified both Sd and Yki in Nerfin-1 immunoprecipitates (*Rhee et al., 2014*).

Nerfin-1 was initially identified as a protein that contains three zinc fingers and that is highly expressed in the nervous system (thus the name Nervous Finger) (*Stivers et al., 2000*). Besides Nerfin-1, the *Drosophila* genome also encodes a related protein called Nerfin-2, which shares sequence similarity with Nerfin-1. However, unlike Nerfin-1, which is expressed in imaginal discs, the expression of Nerfin-2 is undetectable in imaginal discs (*Brown et al., 2014*). We therefore focused our analysis on Nerfin-1 unless otherwise stated. To explore the relationship between Nerfin-1, Sd and Yki, we expressed epitope-tagged constructs for the three proteins in *Drosophila* S2R+ cells and examined their interactions by coimmunoprecipitation (co-IP) assays. Consistent with the DroID data, Sd robustly interacted with Nerfin-1 or Yki in pairwise co-IP assays (*Figure 1A*). In contrast, only weak interaction was detected between Nerfin-1 and Yki (*Figure 1A*). We then examined how each pairwise interaction was influenced by the co-expression of the third protein. While co-expression of Yki did not affect Sd-Nerfin-1 interaction and co-expression of Nerfin-1 did not affect Sd-Yki interaction, the modest Nerfin-1-Yki interaction was greatly potentiated by co-expression of Sd (*Figure 1A–B*). These findings suggest that Sd may simultaneously bind Nerfin-1 and Yki and thus function as an intermediary protein to bridge Nerfin-1 and Yki in a common protein complex. Consistent with this notion, the modest co-IP between Nerfin-1 and Yki was abolished by RNAi knockdown of Sd (*Figure 1C*), demonstrating that the co-IP between Nerfin-1 and Yki constructs was mediated by endogenous Sd present in S2R+ cells.

After demonstrating physical interactions between Nerfin-1 and the Sd-Yki complex, we wished to determine the functional effect of these interactions on the activity of the Sd-Yki complex. For this purpose, we assayed the transcriptional activity of the Sd-Yki complex using a luciferase reporter carrying multimerized minimal Hpo Responsive Element (HRE) derived from the Hippo target gene *diap1* (*Wu et al., 2008*). As expected, the HRE-luciferase reporter was increased 35-fold by co-expression of Sd and Yki (*Figure 1D*). Interestingly, co-expression of Nerfin-1 repressed the Sd-Yki-stimulated HRE-luciferase reporter close to the basal level (*Figure 1D*), suggesting that Nerfin-1 can antagonize the transcriptional activity of the Sd-Yki complex. To probe the molecular mechanisms underlying Nerfin-1's repressive activity, we searched DroID database for other Nerfin-1-interacting proteins. Interestingly, CtBP, a component of the CtBP corepressor complex that represses gene transcription through the associated histone deacetylase (HDACs, Rpd3 in *Drosophila*), was identified as a Nerfin-1-associated protein in a large-scale IP/MS screen (*Guruharsha et al., 2011*). By co-expressing epitope-tagged CtBP or Rpd3 together with Nerfin-1 in S2R+ cells, we found that both proteins could be immunoprecipitated by Nerfin-1 (*Figure 1E–F*), suggesting that the CtBP complex may be involved in Nerfin-1-mediated transcriptional repression. Consistent with this notion, treating S2R+ cells with the HDAC inhibitor Trichostatin A (TSA) significantly reversed Nerfin-1's inhibition of Sd-Yki-stimulated HRE-luciferase reporter, despite that TSA itself had negligible effect on the basal activity of the HRE-luciferase reporter (*Figure 1D*). Taken together, these findings support a model whereby Nerfin-1 antagonizes the transcriptional activity of the Sd-Yki complex by binding to Sd and recruiting transcriptional corepressors such as the CtBP corepressor complex.

To test whether the function of Nerfin-1 is evolutionarily conserved, we examined Insulinoma-associated 1 (INSM1), a mammalian homologue of Nerfin-1. Similar to their *Drosophila* counterparts,

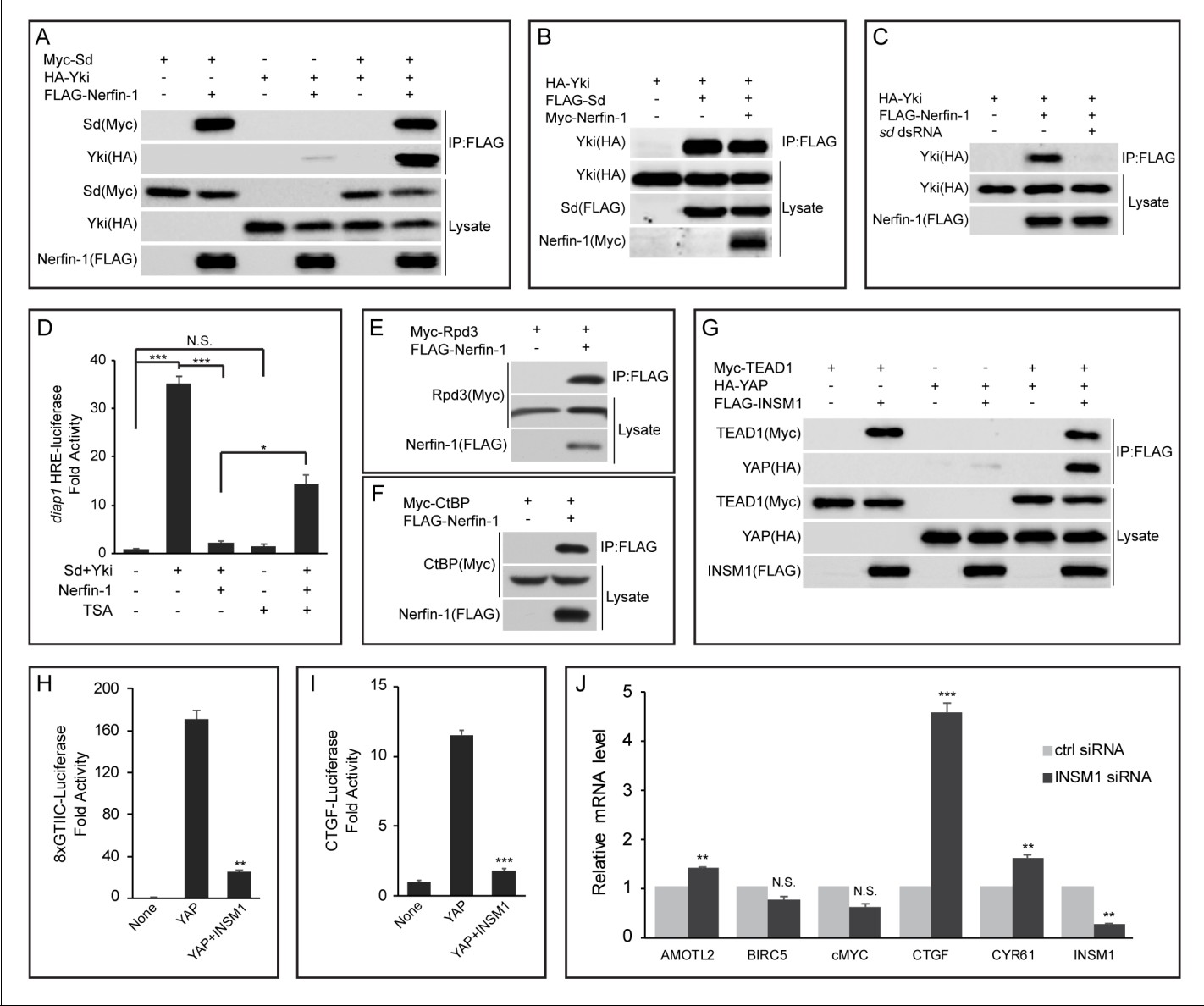

**Figure 1.** Nerfin-1 binds to Sd and antagonizes transcriptional activity of the Sd-Yki complex. (**A**) S2R+ cells expressing the indicated constructs were subjected to IP by anti-FLAG. Note the strong interaction between FLAG-Nerfin-1 and Myc-Sd. Also note the weak interaction between FLAG-Nerfin-1 and HA-Yki, which was strengthened by expression of Sd. (**B**) The indicated constructs were transfected into S2R+ cells and subjected to IP by anti-FLAG. Note that Nerfin-1 did not affect Sd-Yki interaction. (**C**) The indicated constructs and dsRNAs were transfected into S2R+ cells and subjected to IP by anti-FLAG. Note that knockdown of endogenous *sd* abolished Nerfin-1-Yki interaction. (**D**) HRE-luciferase reporter was measured in triplicates in *Drosophila* S2R+ cells transfected with the indicated constructs, with or without the treatment of 200 nM TSA. Note that the activation of HRE-luciferase by Sd and Yki was suppressed by Nerfin-1, and this effect was reversed by TSA. The values are mean ± SEM, *p < 0.05, ***p < 0.001, N.S., non-significant. (**E–F**) Interaction between Nerfin-1 and Rpd3 (**E**) or CtBP (**F**). The indicated constructs were transfected into S2R+ cells and subjected to IP by anti-FLAG. (**G**) 293T cells expressing the indicated constructs were subjected to IP by anti-FLAG. Note the strong interaction between INSM1 and TEAD1. Also note the weak interaction between INSM1 and YAP, which was strengthened by expression of TEAD1. (**H–I**) 8xGTIIC-luciferase (**H**) or CTGF-luciferase (**I**) reporter was measured in triplicates in 293T cells transfected with the indicated constructs. The values are mean ± , **p < 0.01, ***p < 0.001. (**J**) Quantitative real-time PCR analysis of canonical YAP target genes from H727 cells transfected with control siRNA or siRNA against INSM1. Note the upregulation of AMOTL2, CTGF and CYR61. Transfection were performed in quadruplets and values are mean ± SEM, **p < 0.01, ***p < 0.001, N.S., non-significant.

DOI: https://doi.org/10.7554/eLife.38843.003

TEAD1 and YAP were immunoprecipitated by INSM1 in 293T cells (*Figure 1G*). Further echoing our findings in *Drosophila*, YAP-INSM1 interaction was much weaker than TEAD1-INSM1 interaction, and co-expression of TEAD1 significantly enhanced YAP-INSM1 (*Figure 1G*). Indeed, INSM1 potently suppressed YAP-induced transcriptional activation of luciferase reporters driven by multimerized TEAD binding sites (8xGTIIC-luc) (*Dupont et al., 2011*) or CTGF promoter (CTGF-luc) (*Zhao et al., 2008*) (*Figure 1H–I*). To further examine its role in transcriptional regulation, we measured the mRNA levels of a common set of YAP target genes in a lung carcinoid tumor cell line (H727) with high endogenous INSM1 expression. In agreement with our luciferase reporter assay, siRNA knockdown of INSM1 resulted in the upregulation of several YAP target genes, most notably CTGF (*Figure 1J*). These results suggest a conserved role for INSM1 in regulating the transcriptional output of Hippo signaling in mammalian cells.

## Genetic interactions between Nerfin-1, Sd and Yki

To further corroborate our model that Nerfin-1 antagonizes the transcriptional output of the Sd-Yki complex by binding to Sd, we assayed Nerfin-1 activity *in vivo*. Consistent with its inhibitory activity on the Sd-Yki transcription complex in S2R+ cells, MARCM (mosaic analysis with a repressible marker) clones with Nerfin-1 overexpression were much smaller than control clones in the eye imaginal discs (*Figure 2A–B*). Importantly, this small-clone phenotype was completely rescued by removal of Sd, demonstrating that Nerfin-1's growth inhibitory activity is dependent on Sd (*Figure 2C–D*). Further supporting a link between Nerfin-1 and Hippo signaling, expression of Nerfin-1 by the *engrailed*-Gal4 driver resulted in smaller posterior compartment in the wing discs accompanied by decreased expression of the Hippo pathway reporter *expanded-lacZ* (*Figure 2E–F*).

That Nerfin-1 antagonizes the activity of the Sd-Yki complex was further supported by genetic interactions among these genes *in vivo*. Overexpression of UAS-Nerfin-1 by the GMR-Gal4 driver resulted in 100% lethality at late pupal stage (*Figure 2G–H*). When Nerfin-1 was co-expressed with Yki by the GMR-Gal4 driver, we observed a mutual suppression: on the one hand, the lethality of GMR>Nerfin-1 flies was suppressed by Yki co-expression to 55% (*Figure 2M*); on the other hand, Nerfin-1 overexpression significantly suppressed the enlarged eye size resulting from Yki overexpression (*Figure 2I–J*). A similar mutual suppression was observed in flies expressing Nerfin-1, Sd and Yki: Nerfin-1 overexpression significantly suppressed the enlarged eye size resulting from Sd-Yki overexpression (*Figure 2K–L*). Consistent with our MARCM analysis showing that Nerfin-1's growth inhibitory activity is dependent on Sd, halving the dosage of endogenous *sd* dominantly rescued the GMR>Nerfin-1 flies from 100% pupal lethality to 47% survival to adulthood (*Figure 2M*). These dosage-sensitive genetic interactions further support our model implicating Nerfin-1 as a transcriptional repressor that impinges on the Sd-Yki complex.

## The TEA DNA-binding domain of Sd is required for Nerfin-1-binding and transcriptional repression

The data presented so far suggest that Nerfin-1 can inhibit the activity of the Sd-Yki complex by binding to Sd. Furthermore, Nerfin-1 and Yki do not compete with each other for Sd-binding, and the three proteins can co-exist in a trimeric protein complex. The simplest model to account for these findings is that Nerfin-1 and Yki non-competitively bind to different regions of Sd, with Nerfin-1 repressing while Yki activating the transcriptional outcome of Sd. We tested this model by mapping the domains in Sd and Nerfin-1 that are required for Sd-Nerfin-1 interactions.

Sd contains two previously characterized domains, an N-terminal TEA domain that is known to bind DNA and a C-terminal domain that is known to bind Yki. Unlike full-length Sd, a truncated Sd protein lacking the N-terminal TEA domain did not interact with Nerfin-1 in pairwise co-IP assays, and this mutant also lost the ability to enhance the interactions between Nerfin-1 and Yki (*Figure 3A–B*). These results suggest that the TEA domain of Sd is required for binding to Nerfin-1.

In a previous study, X-ray structure of the TEA domain of human TEF-1 revealed a three-helix bundle structure in which helix 3 (H3) recognizes DNA while a hydrophobic surface formed by helix 1 (H1) and helix 2 (H2) was speculated to provide a docking surface for unknown protein(s) (*Anbanandam et al., 2006*) (*Figure 3—figure supplement 1A–B*). We tested whether this hypothetical protein docking surface is required for Nerfin-1-binding by mutating two conserved hydrophobic residues on this surface, Y108 and L130 (*Figure 3—figure supplement 1A–B*). Similar to truncated

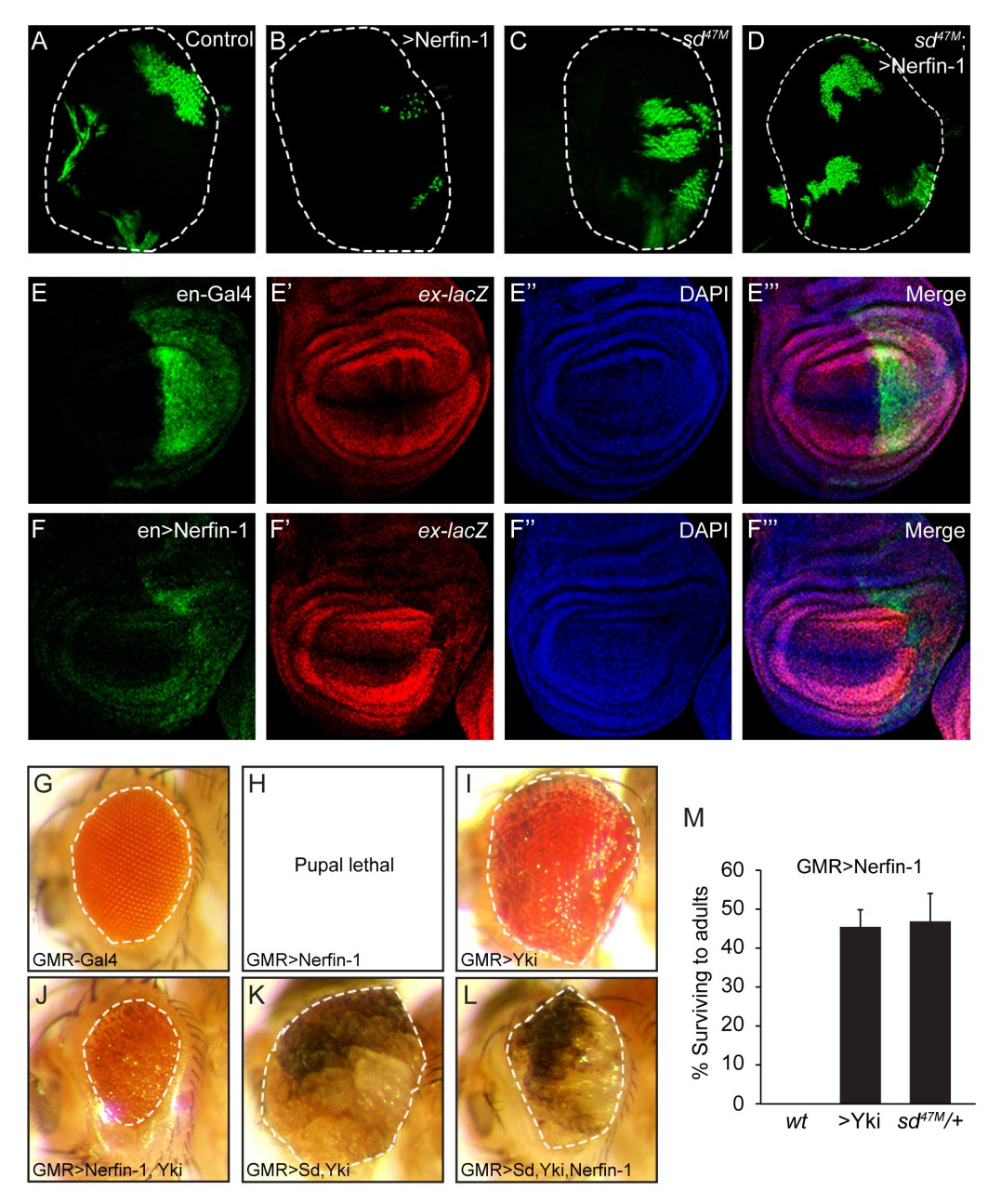

**Figure 2.** Genetic interactions between Nerfin-1, Sd and Yki. (A–D) Eye discs containing GFP-marked MARCM clones of wildtype control (A), Nerfin-1 overexpression (B), $sd^{47M}$ mutant (C) or $sd^{47M}$ mutant with Nerfin-1 overexpression (D). Note the small size of Nerfin-1-overexpression clones and the normal size of $sd^{47M}$ mutant clones with Nerfin-1 overexpression. (E–F''') Wing discs expressing UAS-GFP only (E–E''') or UAS-GFP plus UAS-Nerfin-1 (F–F''') in the posterior compartment by the *en*-Gal4 driver. Note the reduced size of the posterior compartment and reduced expression of *ex-lacZ* upon Nerfin-1 overexpression (compare E-E''' and F-F'''). (G–L) Adult eyes from flies overexpressing the indicated genes by the GMR-Gal4 driver, taken under the same magnification. (M) The percentage of GMR>Nerfin-1 flies surviving to adults was quantified relative to the expected number in the indicated genetic background (mean ± SEM). Three independent crosses were performed for each genotype. The complete genotypes are: (wt) GMR-Gal4/+; UAS-Nerfin-1/+; (>Yki) GMR-Gal4 UAS-Yki/+; UAS-Nerfin-1/+; ($sd^{47M}$/+) $sd^{47M}$/+; GMR-Gal4/+; UAS-Nerfin-1/+.

DOI: https://doi.org/10.7554/eLife.38843.004

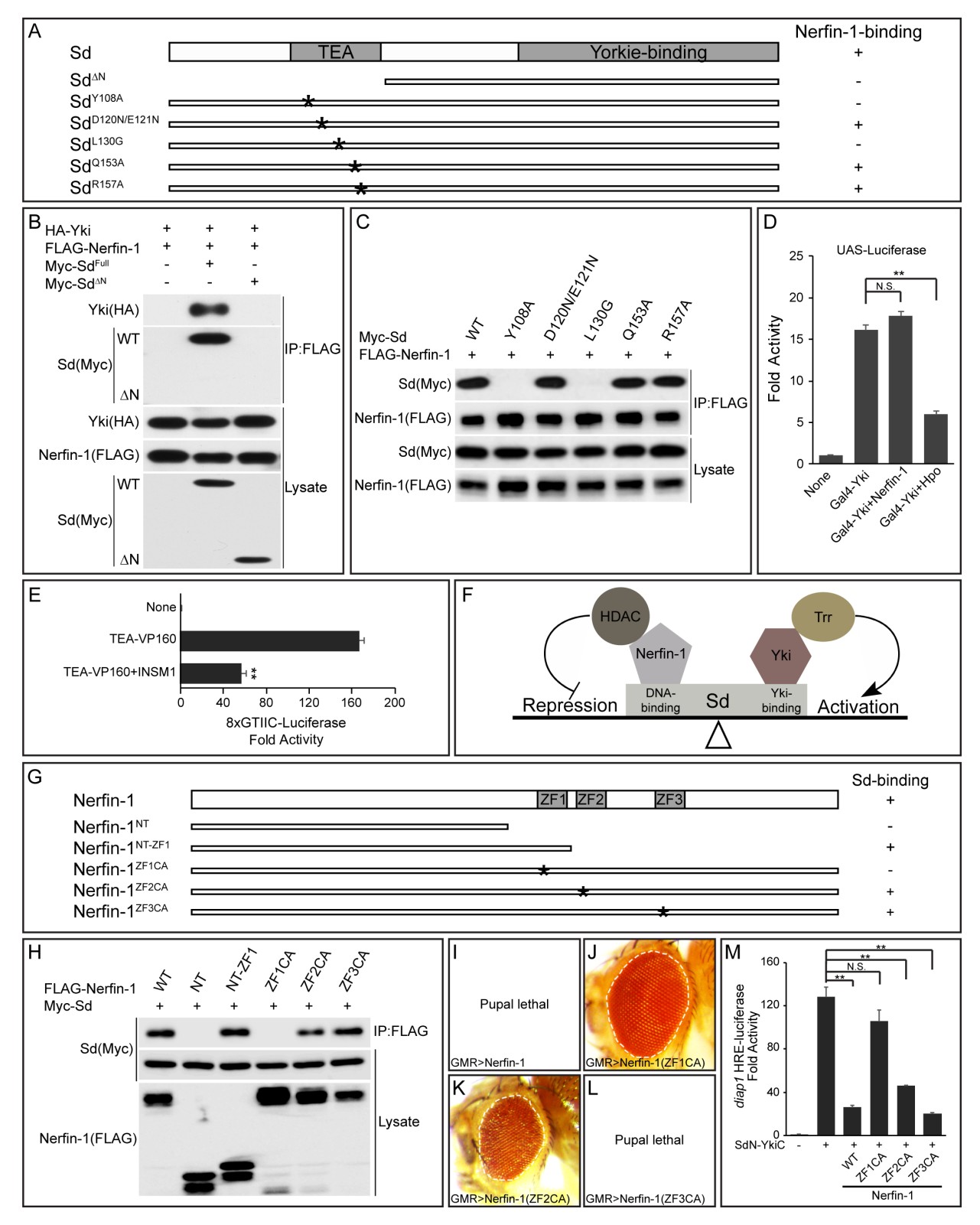

**Figure 3.** The TEA DNA-binding domain of Sd and the zinc fingers of Nerfin-1 are required for Sd-Nerfin-1 interaction. (**A**) A summary of Sd mutants and their interactions with Nerfin-1. The domain structure of Sd and the location of the point mutations tested are also shown. (**B–C**) S2R+ cells expressing the indicated constructs were subjected to IP by anti-FLAG. Sd-Nerfin-1 interaction was abolished by deletion of the TEA domain (ΔN in B) or point mutations on the predicted hydrophobic surface of the TEA domain (Y108A and L130G in C), but not other point mutations in the TEA domain
*Figure 3 continued on next page*

*Figure 3 continued*

(D120N/E121N, Q153A and R157A in C). See *Figure 3—figure supplement 1A–B* for a detailed description of the point mutants tested. (D) UAS-luciferase reporter was measured in triplicates in S2R+ cells expressing the indicated constructs. Note the suppression of Gal4-Yki activity by Hpo, but not Nerfin-1. The values are mean ± SEM, **p < 0.01, N.S., non-significant. (E) 8xGTIIC-luciferase reporter was measured in triplicates in 293T cells expressing the indicated constructs. Note the repression of TEA-VP160 by INSM1. The values are mean ± SEM, **p < 0.01. (F) A model depicting the opposing activities of Nerfin-1 and Yki in regulating the transcriptional output of Sd, through the recruitment of enzymes conferring repressive (HDAC) and active (Trr) histone modification, respectively. (G) A summary of Nerfin-1 mutants and their interactions with Sd. The domain structure of Nerfin-1 and the location of the point mutations tested are also shown. (H) S2R+ cells expressing the indicated constructs were subjected to IP by anti-FLAG. Note that Sd-Nerfin-1 interaction was strongly abolished by mutation of ZF1 (ZF1CA), weakly abolished by mutation of ZF2 (ZF2CA), but not affected by mutation of ZF3 (ZF3CA). (I–L) Adult eyes from flies expressing the indicated Nerfin-1 mutants by the GMR-Gal4 driver. (M) HRE-luciferase reporter was measured in triplicates in S2R+ cells expressing the indicated constructs. The values are mean ± SEM, **p < 0.01, N.S., non-significant.

DOI: https://doi.org/10.7554/eLife.38843.005

The following figure supplements are available for figure 3:

**Figure supplement 1.** Structure-functional analysis of Sd and Nerfin-1.

DOI: https://doi.org/10.7554/eLife.38843.006

**Figure supplement 2.** Nerfin-1 binds to the *diap1* HRE site and impairs Sd-DNA binding.

DOI: https://doi.org/10.7554/eLife.38843.007

Sd lacking the entire TEA domain, an $Sd^{Y108A}$ or $Sd^{L130G}$ mutation also abolished Nerfin-1-Sd interaction (*Figure 3C*). In contrast, mutation of residues in the loop linking H1 and H2 ($Sd^{D120N/E121N}$), or residues in the DNA-contacting H3 ($Sd^{Q153A}$ and $Sd^{R157A}$), had no effect on Nerfin-1-Sd interactions (*Figure 3C*). Thus, Nerfin-1 represents a strong candidate for the unknown protein that was previously speculated to bind to the hydrophobic surface on the TEA domain (*Anbanandam et al., 2006*).

To further corroborate the functional significance of Nerfin-1-Sd binding in inhibiting the transcriptional output of the Sd-Yki complex, we took advantage of a fusion protein between the DNA-binding domain of Gal4 and Yki (Gal4-Yki) (*Huang et al., 2005*). Since Gal4-Yki can directly activate a UAS-luciferase reporter in an Sd-independent manner, one would expect Gal4-Yki to be immune to Nerfin1-mediated repression. Indeed, despite Nerfin-1's potent inhibitory activity on Sd-Yki-stimulated HRE-luciferase reporter (*Figure 1D*), Nerfin-1 was completely inactive towards the transcriptional activity of Gal4-Yki (*Figure 3D*). In contrast, as reported before (*Huang et al., 2005*), Gal4-Yki was significantly inhibited by co-expression of the upstream tumor suppressor Hpo (*Figure 3D*). These results provide further support for our model that Nerfin-1 binding to the TEA domain of Sd is required for Nerfin-1-mediated transcriptional repression.

As an additional test of the importance of Sd's TEA domain in Nerfin-1-mediated transcriptional repression, we engineered a fusion protein containing the N-terminal TEA domain of Sd and the C-terminal transactivation domain of Yki. As expected, the SdN-YkiC fusion protein potently activated the HRE-luciferase reporter (*Figure 3M*). Importantly, unlike the Gal4-Yki fusion protein, Nerfin-1 significantly inhibited the transcriptional activity of the SdN-YkiC fusion protein (*Figure 3M*). To test whether human INSM1 also executes its repressor function through the TEA domain of TEAD, we engineered a fusion protein containing the TEA domain of TEAD1 and the VP160 transcription activation domain. As expected, INSM1 also repressed the transcriptional activity of the TEA-VP160 fusion protein, as measured by the TEAD-responsive 8xGTIIC-luciferase reporter (*Figure 3E*). Taken together, these findings highlight the TEA domain of Sd/TEAD as the molecular target of Nerfin-1-mediated transcriptional repression (*Figure 3F*).

## The zinc fingers of Nerfin-1 are required for Sd-binding and transcriptional repression

The most prominent structural feature of Nerfin-1 is the presence of three zinc fingers, referred to as ZF1, ZF2 and ZF3 hereafter (*Figure 3—figure supplement 1C–D*). To examine whether the zinc fingers are required for Sd-binding, we generated C-terminal truncations lacking all or some of the ZFs, as well as full-length Nerfin-1 carrying Cys-to-Ala mutations of two conserved cystine residues in each ZF (ZF1CA, ZF2CA or ZF3CA) (*Figure 3G* and *Figure 3—figure supplement 1D*). In co-IP assays, a truncated Nerfin-1 protein lacking all ZFs (NT) completely abolished Nerfin-1-Sd interactions, while a truncated protein retaining ZF1 (NT-ZF1) could still bind Sd, suggesting that ZF1 is

required for Nerfin-1-Sd interaction (*Figure 3G–H*). Consistent with this notion, the ZF1CA but not ZF3CA mutation abolished Nerfin-1-Sd interaction, while the ZF2CA mutation partially reduced Nerfin-1-Sd interaction (*Figure 3G–H*).

Next, we assayed the activity of the Nerfin-1 mutants *in vivo*. While overexpression of wildtype Nerfin-1 or ZF3CA by the GMR-Gal4 driver resulted in 100% pupal lethality, overexpression of ZF1CA by the same driver resulted in 100% flies surviving to adulthood and these flies had normal eye size (*Figure 3I–J and L*). Overexpression of ZF2CA had an intermediate effect, with 25% flies surviving to adulthood with reduced eye size (*Figure 3K*). A similar trend was observed in HRE-luciferase reporter assay: ZF1CA was inactive and ZF3CA was fully functional in inhibiting the transcriptional activity of the SdN-YkiC fusion protein, while ZF2CA was partially functional (*Figure 3M*). These structure-functional analyses further support our model implicating Nerfin-1 as a Sd-binding transcriptional repressor.

To further investigate the molecular mechanisms by which Nerfin-1 represses Sd-mediated transcription, we performed chromatin immunoprecipitation (ChIP) to examine the physical interactions between these proteins and the endogenous *diap1* HRE site in S2R+ cells (*Figure 3—figure supplement 2*). As expected, both Sd and Nerfin-1 associated with the *diap1* HRE. Interestingly, the binding of Sd to the HRE was decreased by the co-expression of Nerfin-1 but not a Nerfin-1 mutant protein defective in Sd-binding (ZF1CA). Together with our data implicating CtBP/HDAC in Nerfin-1 function (*Figure 1D–F*), these results suggest that Nerfin-1 represses Sd-dependent transcription both by compromising Sd-DNA interaction and by recruiting CtBP/HDAC corepressors.

## Nerfin-1 is dispensable for Hippo signaling in normal development

To investigate the physiological requirement of Nerfin-1, we examined loss-of-function mutant of *nerfin-1*. It was reported before that *nerfin-1* is an essential gene, and homozygous mutant animals die at late embryonic stage (*Kuzin et al., 2005*). We therefore examined *nerfin-1* mutant clones in several developmental contexts that are known to involve Hippo signaling. Given the antagonistic relationship between Nerfin-1 and Yki in dictating the transcriptional output of Sd, we expected that removal of endogenous Nerfin-1 may result in phenotypes resembling those caused by elevated Yki activity. Unexpectedly, we failed to detect such mutant phenotypes in multiple developmental contexts. First, mosaic eyes containing mutant clones for a null allele of *nerfin-1* did not show increased representation of mutant tissues (*Figure 4A–B*). Second, *nerfin-1* mutant clones in the pupal retina contained normal number of interommatidial cells, which is known to be exquisitely sensitive to Yki activity (*Figure 4—figure supplement 1A–B*). Third, *nerfin-1* mutant cells in the eye or the wing imaginal discs showed normal expression of the Hippo pathway targets Diap1, Expanded and Merlin (Figure 6A–B and *Figure 4—figure supplement 1E–F*). Fourth, posterior follicle cells mutant for *nerfin-1* in stage seven egg chambers did not show elevated expression of Cut, a hallmark of activated Yki (*Figure 4—figure supplement 1G*). Suspecting that loss of Nerfin-1 may be compensated for by the related Nerfin-2 protein, we generated a predicted null mutant allele of *nerfin-2* by CRISPR/Cas9 (*Figure 4—figure supplement 1H*). *nerfin-2* is a non-essential gene, as *nerfin-2* homozygotes were fully viable without detectable abnormalities. Like *nerfin-1* mutant clones, double mutant clones lacking both *nerfin-1* and *nerfin-2* still contained normal number of interommatidial cells in pupal retina (*Figure 4—figure supplement 1C–D*). Thus, at least in the developmental contexts we have examined, loss of Nerfin-1 does not lead to visible phenotypes resembling those caused by increased Yki activity.

## Nerfin-1 suppresses winner cell advantages in cell competition

The apparent dispensability of Nerfin-1 in normal development promoted us to examine its requirement in other biological processes that involve Hippo signaling. Recent studies in both *Drosophila* and mammalian cells have implicated the Hippo pathway in cell competition: not only does high Yki/YAP activity lead to a winner cell fate, it also protects loser cells from being eliminated in *Minute*-mediated cell competition (*Neto-Silva et al., 2010*; *Tyler et al., 2007*; *Ziosi et al., 2010*). We therefore examined whether Nerfin-1 may be required for the regulation of Yki activity in cell competition.

We first used eyeless-Flp to generate adult mosaic eyes containing $Rps17^4$ heterozygous *Minute* cells labeled by red pigmentation and white-colored wildtype cells lacking red pigmentation. Thus,

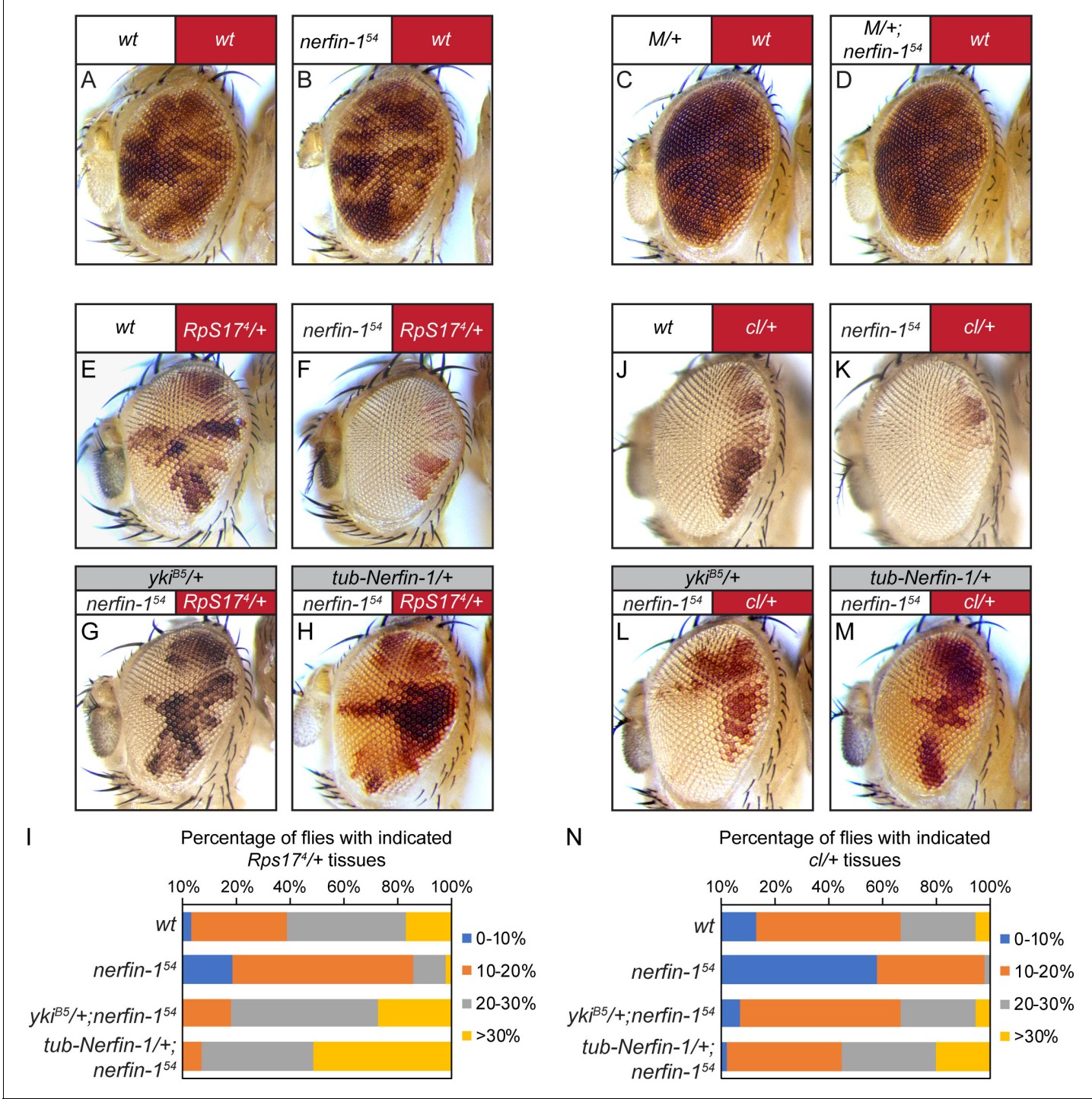

**Figure 4.** Loss-of-*nerfin-1* in the winner cells enhances cell competition. In all images of mosaic eyes, the genotypes of the constituting cells are indicated with matching colors (red or white) above the eye image. (**A–B**) Mosaic eyes containing control clones (**A**) or *nerfin-1⁵⁴* mutant clones (**B**). Note the similar representation of *nerfin-1* mutant clones compared to control clones. The complete genotypes are: (**A**) *y w ey-Flp; FRT80B/P[w⁺] FRT80B*; (**B**) *y w ey-Flp; nerfin-1⁵⁴ FRT80B/P[w⁺] FRT80B*. (**C–D**) Mosaic eyes containing *Minute* clones (**C**) or *nerfin-1⁵⁴* mutant *Minute* clones (**D**). Note the similar representation of *nerfin-1* mutant *Minute* clones compared to control *Minute* clones. The complete genotypes are: (**C**) *y w ey-Flp/Df(1) R194 w; FRT80B/P[RpL36⁺w⁺] FRT80B*; (**D**) *y w ey-Flp/Df(1)R194 w; nerfin-1⁵⁴ FRT80B/P[RpL36⁺w⁺] FRT80B*. See **Figure 4—figure supplement 2E** for a detailed description of the method to generate mosaic eyes with loss-of-*nerfin-1* specifically in *Minute* cells. (**E–I**) Mosaic eyes of the indicated genotypes (**E–H**). The percentage of red-colored *Minute* tissue in each mosaic eye was measured and assigned to one of the following four groups: 0–10%, 10–20%, 20–30% and >30%. The number of flies in each group was counted and the percentage of each group was represented in bins in (**I**). The

*Figure 4 continued on next page*

*Figure 4 continued*

complete genotypes are: (E) *y w ey-Flp; FRT80B/P[w⁺] Rps17⁴ FRT80B* (n = 105); (F) *y w ey-Flp; nerfin-1⁵⁴ FRT80B/P[w⁺] Rps17⁴ FRT80B* (n = 101); (G) *y w ey-Flp; yki^{B5}/+; nerfin-1⁵⁴ FRT80B/P[w⁺] Rps17⁴ FRT80B* (n = 105); (H) *y w ey-Flp; tub-Nerfin-1/+; nerfin-1⁵⁴ FRT80B/P[w⁺] Rps17⁴ FRT80B* (n = 124). (J–N) Mosaic eyes of the indicated genotype (J–M). The percentage of red-colored tissue heterozygous for cell-lethal mutation in each mosaic eye was measured and assigned to one of the following four groups: 0–10%, 10–20%, 20–30% and >30%. The number of flies in each group was counted and the percentage of each group was represented in bins in (N). The complete genotypes are: (J) *y w ey-Flp; FRT80B/P[w⁺] l(3)CL-L¹ FRT80B* (n = 68); (K) *y w ey-Flp; nerfin-1⁵⁴ FRT80B/P[w⁺] l(3)CL-L¹ FRT80B* (n = 65); (L) *y w ey-Flp; yki^{B5}/+; nerfin-1⁵⁴ FRT80B/P[w⁺] l(3)CL-L¹ FRT80B* (n = 115); (M) *y w ey-Flp; tub-Nerfin-1/+; nerfin-1⁵⁴ FRT80B/P[w⁺] l(3)CL-L¹ FRT80B* (n = 151).

DOI: https://doi.org/10.7554/eLife.38843.008

The following figure supplements are available for figure 4:

**Figure supplement 1.** Nerfin-1 is dispensable for imaginal disc growth and follicle cell differentiation.

DOI: https://doi.org/10.7554/eLife.38843.009

**Figure supplement 2.** Loss-of-*nerfin-1* in the winner cells enhances cell competition.

DOI: https://doi.org/10.7554/eLife.38843.010

the relative representation of red-colored *Minute* tissue and white-colored wildtype tissue in the adult eyes provides an easy readout for cell competition between the loser (*Minute*) and the winner (wildtype) cells (*Figure 4E*; quantified in *Figure 4I*). Interestingly, when compared to mosaic eyes containing wildtype winner cells (*Figure 4E*), mosaic eyes in which the winner cells were mutant for *nerfin-1* had significantly lower percentage of red tissue and significantly higher percentage of white tissue (*Figure 4F*; quantified in *Figure 4I*), suggesting that loss-of-*nerfin-1* enhanced the ability of winner cells to eliminate loser cells in *Minute*-mediated cell competition.

For simplicity, we will use the term 'super-winner' to describe the ability of loss-of-*nerfin-1* to further enhance the advantage of wildtype cells when juxtaposed with less fit cells, in contrast to the term 'super-competitor', which describes cells with higher fitness than wildtype cells (*Moreno and Basler, 2004*). Interestingly, halving the dosage of endogenous *yki* was sufficient to reverse this 'super-winner' phenotype conferred by loss-of-*nerfin-1* (*Figure 4G*; quantified in *Figure 4I*), consistent with our model implicating Nerfin-1 as an antagonist of Yki. Besides *Rps17*, loss-of-*nerfin-1* conferred a similar 'super-winner' phenotype when mutation of another ribosomal subunit *RpL14* was used to generate *Minute* tissues in the eyes (*Figure 4—figure supplement 2A–D*). To ensure that the 'super-winner' phenotype was caused by loss of *nerfin-1*, we performed a rescue experiment and found that the 'super-winner' phenotype was rescued by a *tubulin-nerfin-1* transgene (*Figure 4H*; quantified in *Figure 4I*).

Next, we took advantage of a genetic setup that enabled us to mutate *nerfin-1* specifically in the loser cells in the context of *Minute*-mediated competition (*Figure 4—figure supplement 2E*) (*Tyler et al., 2007*). In contrast to loss-of-*nerfin-1* in the winner cells, loss-of-*nerfin-1* in the loser cells had no effect on the representation of winner vs. loser tissues in adult eyes (*Figure 4C–D*). Taken together, these results suggest that Nerfin-1 is preferentially required in the winner cells to suppress *Minute*-mediated cell competition.

After demonstrating the requirement of Nerfin-1 in suppressing winner cell advantages in *Minute*-mediated cell competition, we extended our analysis to another condition that involves the juxtaposition of cells with different fitness (*Hafezi et al., 2012*). Instead of competition between *Minute* (red) and wildtype (white) cells, we assayed interactions between cells heterozygous for a recessive cell-lethal mutation (red) and wildtype (white) cells (*Figure 4J*; quantified in *Figure 4N*). Interestingly, analogous to what we have observed in *Minute*-mediated cell competition, loss-of-*nerfin-1* increased the representation of wildtype cells relative to cells heterozygous for the cell-lethal mutation and halving the dosage of endogenous *yki* similarly reversed this phenotype conferred by loss-of-*nerfin-1* (*Figure 4K–L*; quantified in *Figure 4N*). Furthermore, the increased representation of *nerfin-1* mutant cells in this context was also rescued by a *tubulin-nerfin-1* transgene (*Figure 4M*; quantified in *Figure 4N*). Taken together, we conclude that Nerfin-1 suppresses winner cell advantages in *Minute*- and cell lethal-mediated cell competition, both in a Yki-dependent manner. Importantly, in both contexts, the overall eye size was not changed, suggesting that the observed changes in red/white ratio results from modulation of cell competition, not simply tissue overgrowth.

To trace the developmental origin of the 'super-winner' phenotype conferred by loss-of-*nerfin-1*, we used eyeless-Flp to generate 3ʳᵈ instar mosaic eyes containing *Rps17⁴* heterozygous *Minute* cells

that were positively labeled by β-galactosidase. We then followed cell death, a hallmark of loser cell fate in cell competition (*Moreno et al., 2002*), by staining for active effector caspase Dcp-1. Consistent with results from the adult eyes, compared to mosaic eyes containing wildtype winner cells, we observed a 50% decrease of *Minute* tissues (β-gal-positive) when winner cells were mutant for *nerfin-1* (*Figure 5A–C*). Concomitantly, we saw a two-fold increase of apoptotic *Minute* cells (cells positive for both β-gal and Dcp-1) when winner cells were mutant for *nerfin-1* (*Figure 5D*). Next, we repeated the same analysis in mosaic wing discs containing $Rps17^4/+Minute$ cells. As in the eye discs, we observed a similar decrease of *Minute* tissues in the wing discs accompanied by an increase of apoptotic *Minute* cells when winner cells were mutant for *nerfin-1* (*Figure 5E–H*). These data further support our model that the Nerfin-1 is required in the winner cells to suppress *Minute*-mediated cell competition.

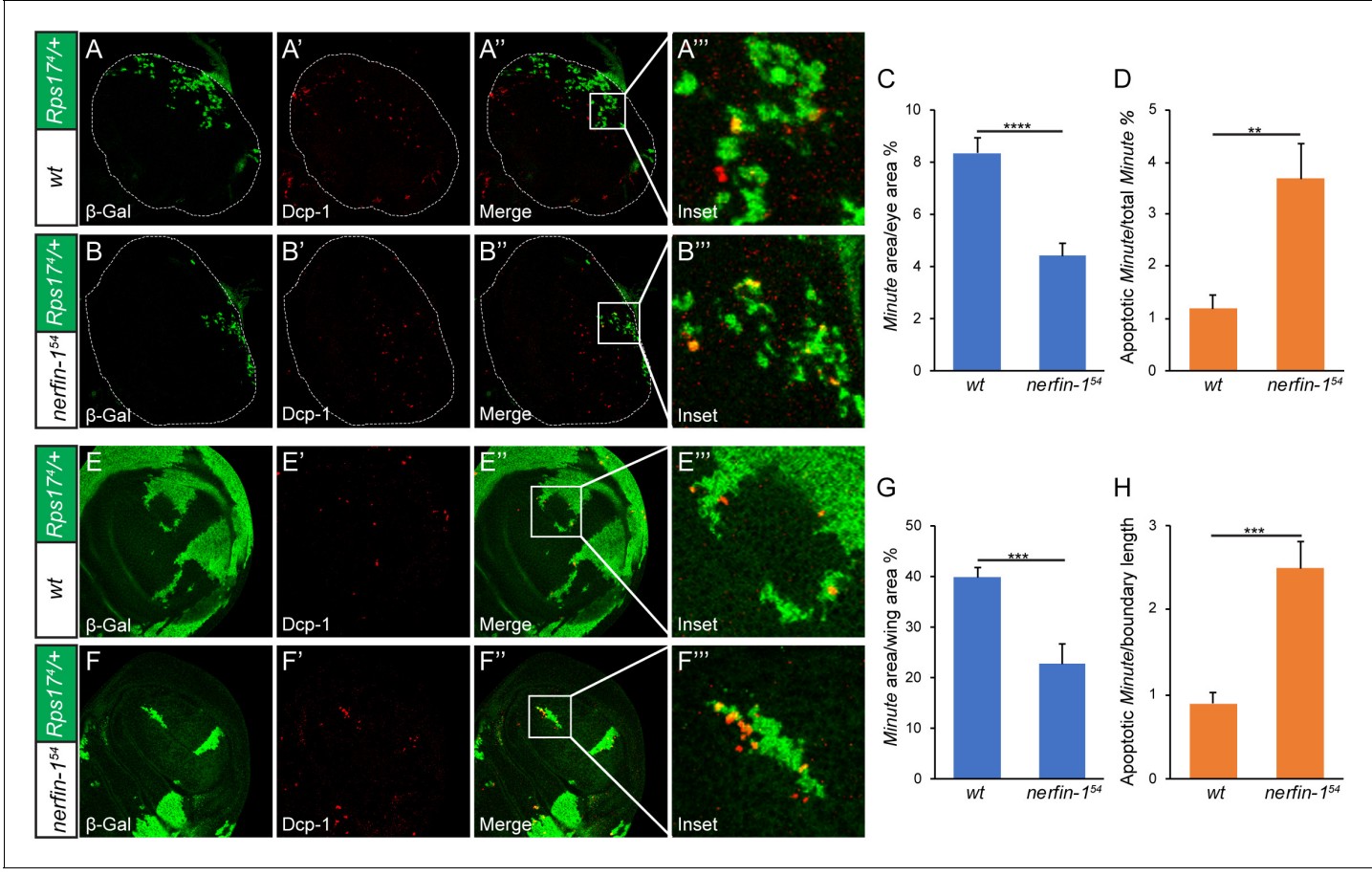

**Figure 5.** Developmental origin of the 'super-winner' phenotype conferred by loss-of-*nerfin-1*. (A–B''') Mosaic larval eye discs of the indicated genotypes were stained for β-Gal (green; marking the *Minute* loser cells) and Dcp-1 (red). Enlarged views of the boxed area in (A'') and (B'') are shown in A''' and B''', respectively. The complete genotypes are: (A) *y w ey-Flp; FRT80B/P[arm-lacZ w⁺] Rps17⁴ FRT80B*; (B) *y w ey-Flp; nerfin-1⁵⁴ FRT80B/P[arm-lacZ w⁺] Rps17⁴ FRT80B*. (C) The percentage of *Minute* tissues in eye discs of the indicated genotypes in (A) and (B) was quantified (mean ± SEM, n = 15, ****p < 0.0001. Loss-of-*nerfin-1* decreased the percentage of *Minute* tissues. (D) Quantification of apoptotic loser cells in eye discs of the indicated genotypes in (A) and (B). The number of cells positive for both Dcp-1 and β-Gal (indicating *Minute* cells undergoing apoptosis) relative to all β-Gal-positive cells (indicating all *Minute* cells) was plotted. The values are mean ± SEM, n = 15, **p < 0.01. (E–F''') Mosaic wing discs of the indicated genotypes were stained for β-Gal (green; marking the *Minute* loser cells) and Dcp-1 (red). Enlarged views of the boxed area in (E'') and (F'') are shown in E''' and F''', respectively. The complete genotypes are: (E) *y w hs-Flp; FRT80B/P[arm-lacZ w⁺] Rps17⁴ FRT80B*; (F) *y w hs-Flp; nerfin-1⁵⁴ FRT80B/P[arm-lacZ w⁺] Rps17⁴ FRT80B*. (G) The percentage of *Minute* tissues in wing discs of the indicated genotypes in (E) and (F) was quantified (mean ± SEM, n = 15, ***p < 0.001. (H) Quantification of apoptotic loser cells in wing discs of the indicated genotypes in (E) and (F). The number of cells positive for both Dcp-1 and β-Gal (indicating *Minute* cells undergoing apoptosis) was quantified per micron of boundary between the winner and loser cells, as described previously (*Li and Baker, 2007*). The values are mean ± SEM, n = 15, ***p < 0.001.

DOI: https://doi.org/10.7554/eLife.38843.011

To further characterize the *nerfin-1*-dependent 'super-winner' phenotype in *Minute*-mediated cell competition, we examined the expression of the Yki target gene *diap1*. In mosaic wing discs containing $Rps17^4/+$ *Minute* cells and wildtype cells, we observed a modest increase of Diap1 expression in the winner cells relative to the loser cells, especially when comparing winner and loser cells immediately abutting the clonal boundary (*Figure 6C*). This observation is consistent with previous reports implicating Hippo signaling in cell competition. Interestingly, when we examined mosaic wing discs containing $Rps17^4/+$ *Minute* loser cells and *nerfin-1* mutant winner cells, a more dramatic difference in Diap1 expression was observed across the winner/loser boundary (*Figure 6D*, quantified in *Figure 6E*). These results provide further support for our model that Nerfin-1 suppresses winner cell advantages in cell competition by regulating the transcriptional output of Hippo signaling.

## Discussion

In this study, we present the characterization of Nerfin-1 as a corepressor that binds the TEA domain of Sd and antagonizes Sd-Yki-mediated transcriptional activation. Several lines of evidence support this conclusion. First, Nerfin-1 physically associates with Sd and Yki, and the three proteins can form a trimeric complex. Second, in the eye discs, overexpression of Nerfin-1 suppresses tissue growth in an Sd-dependent manner. Third, Nerfin-1 genetically interacts with *sd* and *yki* in a dosage-dependent manner. Interestingly, the relationship between Nerfin-1 and Sd appears to be evolutionary conserved, since the mammalian counterparts of Nerfin-1, Sd and Yki also associate with each other and cooperatively regulate YAP target gene expression. Along this line, we note that loss of function of the *C. elegans* ortholog of Nerfin-1 and Sd, *egl-46* and *egl-44*, respectively, resulted in similar

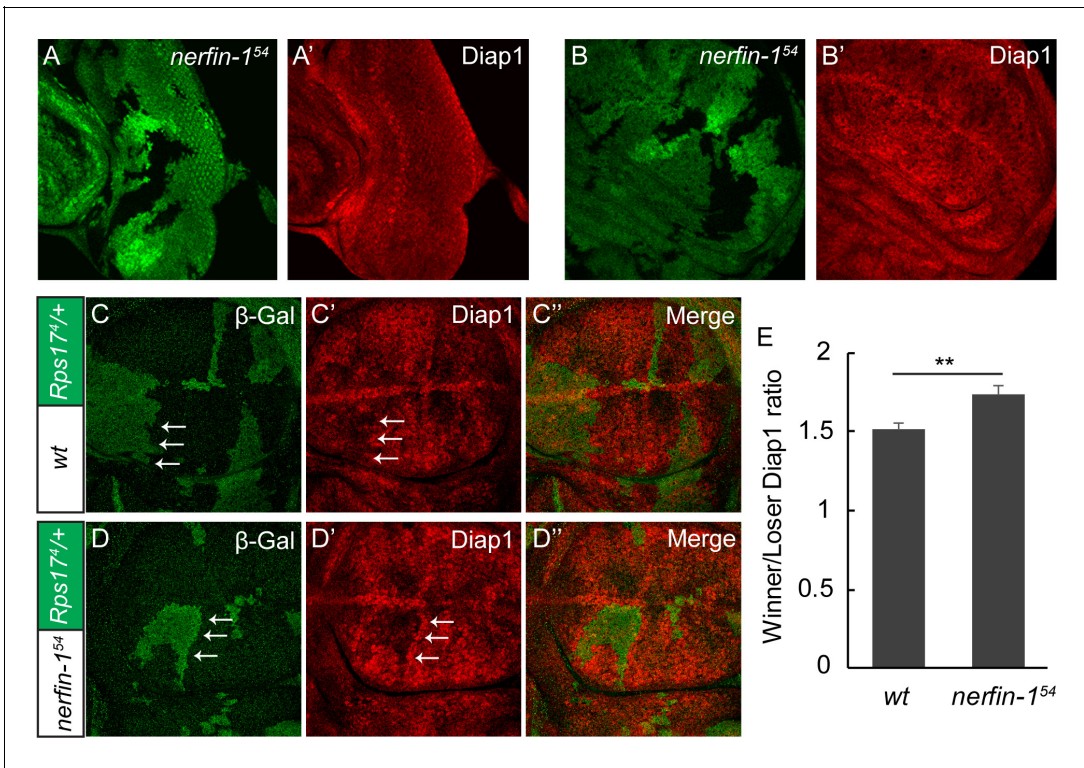

**Figure 6.** Nerfin-1 suppresses the expression of Yki target *diap1* in the winner cells. (A–B') A third instar eye (A–A') or wing disc (B–B') containing GFP-negative *nerfin-1⁵⁴* mutant clones was stained for Diap1 expression. Note the normal expression of Diap1 in mutant clones compared to neighboring wildtype cells. (C–E) Mosaic wing discs of the indicated genotypes were stained for β-Gal (green; marking the *Minute* loser cells) and Diap1 (red). The differential expression of Diap1 across the winner/loser boundary (a representative example of winner/loser boundary is marked by arrows) was quantified in (E) (mean ± SEM, n = 15), **p < 0.01. The complete genotypes are: (C) *y w hs-Flp; FRT80B/P[arm-lacZ w⁺] Rps17⁴ FRT80B*; (D) *y w hs-Flp; nerfin-1⁵⁴ FRT80B/P[arm-lacZ w⁺] Rps17⁴ FRT80B*.
DOI: https://doi.org/10.7554/eLife.38843.012

defects in the specification of FLP and HSN neurons (*Wu et al., 2001*). Thus, the functional interactions between Nerfin-1 and Sd may have a deep evolutionary origin.

It is interesting to compare Nerfin-1 with another Sd-binding co-repressor, Tgi. While both proteins confer transcriptional repression by binding to Sd, there are clear differences in their mode of action. While Tgi and Yki bind to the C-terminal domain of Sd in a mutually exclusive manner (*Koontz et al., 2013*), Nerfin-1 binds to the TEA DNA-binding domain of Sd, apparently independent of the binding of Yki to the C-terminal domain of Sd. Thus, Sd-Tgi binding, but not Sd-Nerfin-1 binding, is modulated by the strength of Hippo signaling. We have examined whether these proteins play redundant roles in Sd-mediated repression *in vivo* by generating *nerfin-1 tgi* double mutant combination, and found that loss of *tgi* does not enhance the proliferation of *nerfin-1* mutant tissues, or the 'super-winner' phenotype of *nerfin-1* mutant in cell competition (*Figure 4—figure supplement 2F–H*). These results suggest that Nerfin-1 and Tgi are likely required in different contexts to suppress Sd function, although we cannot exclude the possibility that they are co-required in yet-to-be-identified biological contexts.

At present, how Tgi mediates transcriptional repression, especially the identity of the co-repressors recruited by Tgi to repress target gene transcription, remains unclear (*Koontz et al., 2013*). By contrast, Nerfin-1 appears to repress target gene transcription by recruiting repressive histone modifying proteins such as CtBP-HDAC. Consistent with this notion, HDACs and CoREST, which are components of the CtBP-HDAC corepressor complex, were identified as proteins associated with the mammalian Nerfin-1 homologue INSM1 by IP/MS (*Welcker et al., 2013*). Interestingly, contrary to the Sd co-repressor Nerfin-1, the Sd co-activator Yki is known to confer transcriptional activation by recruiting the Trithorax-related (Trr) histone methyltransferase complex (*Oh et al., 2014*; *Qing et al., 2014*). Thus, the transcriptional output of Sd is dictated by the integration of positive and negative chromatin modifications at the target loci (*Figure 3F*).

Another insight from this study concerns the role of Nerfin-1 in cell competition. Although Nerfin-1 is dispensable for the growth of imaginal discs and ovarian follicle cells, we show that Nerfin-1 normally suppresses winner cell advantage in cell competition. Accordingly, loss of Nerfin-1 specifically in the winner cells confers a 'super-winner' phenotype, resulting in greater elimination of loser cells and increased representation of winner cells in mosaic tissues. Such 'super-winner' phenotype is mediated by increased Yki activity in the *nerfin-1* mutant winner cells, as reflected by both increased expression of the Yki target *diap1* in the winner cells and suppression of the 'super-winner' phenotype by halving the dosage of endogenous *yki*. Together with our molecular characterization of Nerfin-1, these results suggest that Nerfin-1 normally suppresses winner cell advantage by antagonizing the Sd-Yki complex in the winner cells. Since cell competition is conserved in mammals (*Gogna et al., 2015*), it will be interesting to examine whether the mammalian counterpart of Nerfin-1 (INSM1) also plays a conserved role in cell competition.

Our findings uncovering a Yki-dependent requirement for Nerfin-1 in cell competition have several implications. First, although a number of genetic perturbations are known to cause cell competition, to our knowledge, *nerfin-1* is the first example of mutations that do not confer cell competition per se, but instead modulate the degree of cell competition conferred by others. Thus, cell competition is not simply the constitutive outcome of juxtaposition of cells of different fitness; the process itself is subjected to additional regulation. Second, although previous studies have implicated the Hippo signaling pathway in cell competition (*Neto-Silva et al., 2010*; *Tyler et al., 2007*; *Ziosi et al., 2010*), those studies involved conditions in which Yki is massively activated at supraphysiological level. Our study therefore provides the first evidence that Yki is required for cell competition at its endogenous physiological level. Lastly, given that Nerfin-1 is dispensable for the growth of imaginal discs and ovarian follicle cells but is required for antagonizing Yki activity in cell competition, it is possible that Nerfin-1 is preferentially required in physiological contexts that involve interactions between cells with different Yki activity. Besides cell competition, cells with differential Yki activity have been documented in several examples of epithelial regeneration (*Grusche et al., 2011*; *Losick et al., 2013*). It will thus be interesting to examine the requirement of Nerfin-1 in these processes.

# Materials and methods

## Plasmids and antibodies

FLAG-Nerfin-1 construct was generated from cDNA clone LD18634 obtained from *Drosophila* Genomics Resource Center (DGRC). INSM1 expressing construct was generated from ORF clone (OHu02156) purchased from Genscript. Nerfin-1 zinc finger domain mutant constructs were generated by mutating following residues using site-directed mutagenesis: C252 and C255 were mutated to alanine in ZF1CA; C280 and C283 were mutated to alanine in ZF2CA; C336 and C339 were mutated to alanine in ZF3CA. Sd mutant constructs Y108F, D120N/E121N, L130G, Q153A and R157A were generated by site-directed mutagenesis. The following luciferase reporter constructs have been described previously: *diap1* HRE-luciferase (*Wu et al., 2008*), 8xGTIIC-luciferase (*Dupont et al., 2011*), and CTGF-luciferase (*Zhao et al., 2008*).

Primary antibodies used in this study include the following: FLAG and HA (Sigma); Myc (Calbiochem); Diap1 (gift from Bruce Hay); Expanded and Merlin (gift from Richard Fehon); β-galactosidase and Cut (Developmental Studies Hybridoma Bank).

## *Drosophila* genetics

The following flies have been described previously: $yki^{B5}$ and UAS-Yki (*Huang et al., 2005*); $sd^{47M}$, UAS-Sd and the *diap1*-lacZ reporter $th^{j5c8}$ (*Wu et al., 2008*); $nerfin-1^{54}$ (*Kuzin et al., 2005*); *Df(1) R194 w/FM7; P[Rpl36+w+] arm-lacZ FRT80B/TM3* (*Tyler et al., 2007*); $Rps17^4$; $RpL14^1$ and $l(3)CL-L^1$ (Bloomington Stock Center); UAS-Nerfin-1, UAS-Nerfin-1(ZF1CA-ZF3CA), tubulin-Nerfin-1 transgenes were made in this study. $nerfin-2^{m6-8}$ was generated by CRISPR/Cas9-mediated mutagenesis with the facilitation of the following gRNAs selected by CRISPR Optimal Target Finder (http://tools. flycrispr.molbio.wisc.edu/targetFinder/): GGTCTCATCTTCCACGTAGA and TGACTACAATGAG TACGCCA. Mutants were identified by PCR selection and verified by Sanger sequencing. Representative genotypes used for clonal analysis in imaginal discs are as follows:

GFP+ MARCM clones:

*tub-Gal80 hs-FLP FRT19A/FRT19A; UAS-GFP; tub-Gal4*
*tub-Gal80 hs-FLP FRT19A/FRT19A; UAS-GFP/UAS-Nerfin-1; tub-Gal4*
*tub-Gal80 hs-FLP FRT19A/sd⁴⁷ᴹ FRT19A; UAS-GFP; tub-Gal4*
*tub-Gal80 hs-FLP FRT19A/sd⁴⁷ᴹ FRT19A; UAS-GFP/UAS-Nerfin-1; tub-Gal4*

GFP- mutant clones:

*hs-FLP; Ubi-GFP FRT80B/nerfin-1⁵⁴ FRT80B*

## Quantification of cell competition in imaginal discs

To quantify the *Minute* area/disc area ratio (*Figure 5C and G*), the pixels of the β-Gal-positive *Minute* area (green) in each disc was measured by ImageJ and divided by the pixels of the whole eye disc or wing disc. To quantify apoptotic *Minute* cells in eye discs (*Figure 5D*), we first calculated the total number of *Minute* cells in each disc by dividing the pixels of the β-Gal-positive *Minute* area by the average pixel size of each cell (70 pixels/cell). The apoptotic *Minute*/total *Minute* ratio was then calculated by dividing the number of cells positive for both Dcp-1 and β-Gal by the number of total *Minute* cells in each disc. To quantify apoptotic *Minute* cells in wing discs (*Figure 5H*), we calculated the ratio of Dcp-1-positive apoptotic *Minute* cells per micron of clonal boundary between the winner and loser cells, as described previously (*Li and Baker, 2007*).

## Cell cultures, luciferase assay, qRT-PCR and ChIP assay

*Drosophila* S2R+ cells were cultured in Schneider's medium supplemented with 10% fetal bovine serum (FBS) and antibiotics at 25°C. HEK293T and H727 cells were maintained at 37°C in DMEM and RPMI-1640 medium supplemented with 10% FBS and antibiotics, respectively. Plasmid Transfection were performed using Effectene reagent (Qiagen). siRNA transfection was performed using Lipofectamine RNAiMax reagent. ON-TARGET plus siRNA SMARTpools were purchased from Dharmacon.

Luciferase assay was performed as previously described (*Wu et al., 2008*). Briefly, *Drosophila* S2R + or 293T cells were seeded on a 48-well plate. After 24 hr, cells were transfected with the desired Firefly luciferase reporter constructs together with Pol III-Renilla luciferase reporter plasmid (as the

internal control) using Effectene transfection reagent (Promega). The HRE-luciferase reporter contains 24 tandem copies of the 26 bp minimal HRE from the *diap1* gene (*Wu et al., 2008*). Luciferase assay was performed at 24 hr post-transfection using Dual Luciferase Assay system (Promega) following the manufacturer's instructions and a FLUOstar Lumiometer (BMG Lab Technologies).

ChIP assays were performed according to a previously described protocol (*Qing et al., 2014*). Briefly, ~5 × 10$^6$ S2R+ cells were cross-linked with 1% formaldehyde and sonicated to an average fragment size between 200 bp and 500 bp. Two micrograms of primary antibodies and 50 µl of protein G agarose were used in each assay. The immunoprecipitated DNA was quantified using quantitative real-time PCR. All values were normalized to the input. The primers for analyzing the ChIP are provided as follows:

> HRE Forward: ACGAACACGAAGACCAAA
> HRE Reverse: CTCCAAGCCAGTTTGATT

For quantitative RT-PCR, RNA was extracted using Trizol Reagent (Invitrogen) and purified using RNeasy Mini Kit (Qiagen). For each sample, 1 ug RNA was used for reverse transcription using iScript cDNA synthesis kit (BioRad). 100 ng cDNA was used for real-time PCR with SYBR master mix (Bio-Rad). Real-time PCR was performed using CFX96 real-time system (BioRad).

## Acknowledgements

We thank Dr. Richard Fehon for Ex and Mer antibodies, Dr. Ward Odenwald for *nerfin-1*[54] mutant fly, Dr. Nicholas Baker for *RpL36* transgenic fly, and the Bloomington Stock Centers for *Drosophila* stocks. This work was supported in part by grants from the National Institutes of Health (EY015708). DP is an investigator of the Howard Hughes Medical Institute.

## Additional information

### Competing interests

Pengfei Guo: Reviewing editor, *eLife*. The other authors declare that no competing interests exist.

### Funding

| Funder | Grant reference number | Author |
| --- | --- | --- |
| National Institutes of Health | R01 EY015708 | Duojia Pan |
| Howard Hughes Medical Institute | Investigator | Duojia Pan |

The funders had no role in study design, data collection and interpretation, or the decision to submit the work for publication.

### Author contributions

Pengfei Guo, Conceptualization, Formal analysis, Supervision, Funding acquisition, Project administration, Writing—review and editing; Chang-Hyun Lee, Conceptualization, Data curation, Formal analysis, Investigation, Methodology, Writing—original draft; Huiyan Lei, Resources, Formal analysis, Investigation, Methodology; Yonggang Zheng, Formal analysis, Investigation; Katiuska Daniela Pulgar Prieto, Duojia Pan, Resources, Investigation

### Author ORCIDs

Pengfei Guo (iD) http://orcid.org/0000-0002-4618-6803
Duojia Pan (iD) https://orcid.org/0000-0003-2890-4645

### Decision letter and Author response

Decision letter https://doi.org/10.7554/eLife.38843.015
Author response https://doi.org/10.7554/eLife.38843.016

## Additional files

**Supplementary files**
• Transparent reporting form
DOI: https://doi.org/10.7554/eLife.38843.013

**Data availability**

All data generated or analysed during this study are included in the manuscript and supporting files.

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
