## [Decision Letter]

Thank you for submitting your article "Nerfin-1 represses transcriptional output of Hippo signaling in cell competition" for consideration by *eLife*. Your article has been reviewed by three peer reviewers, and the evaluation has been overseen by Utpal Banerjee as the Reviewing Editor and Didier Stainier as the Senior Editor. All individuals involved in review of your submission have agreed to reveal their identity: Eduardo Moreno (Reviewer #1); Kun-Liang Guan (Reviewer #2); Georg Halder (Reviewer #3).

The reviewers have discussed the reviews with one another and the Reviewing Editor has drafted this decision to help you prepare a revised submission.

Summary:

The Hippo pathway is a critical regulator of growth and its mis-regulation, a major cause of oncogenic transformation in multiple tissues. This manuscript identifies Nerfin-1 as a novel regulator of the Hippo pathway that controls this pathway at the level of Sd DNA binding transcription factor and regulates the activity of the Hippo-regulated Yki transcriptional co-activator. Both biochemical and functional data showed that Nerfin-1 directly binds to Sd and represses gene expression. Nerfin-1 overexpression suppresses tissue growth and this effect is dependent on Sd. Surprisingly, Nerfin-1 deletion does not show a dramatic effect of growth in multiple tissues, suggesting it does not have a major role in physiological organ/tissue size control. However and interestingly, the authors showed that Nerfin-1 plays a role in cell competition as Nerfiin-1 inactivation promotes the winner cells to eliminate loser cells. Finally, data are presented to show that the mammalian Nerfin-1 homolog INSM1 similarly binds to and inhibits TEAD dependent transcription.

The data presented are good quality and support the major conclusions in the paper. Although Vg/VGL4 have been shown to inhibit Sd/TEAD dependent transcription, the mechanism of Nerfin-1/INSM1 in transcription repression of Sd/TEAD is different. Therefore, the study by Pan and colleagues has revealed a new mechanism of Hippo pathway transcription regulation and has high potential significance.

Essential revisions:

1) It is unclear that the cell lethality shown in Figure 4J-N is due to cell competition rather than a case of autonomous lethality of the homozygous cells. The efficiency of ey-flp is quite high and therefore, Figure 4E and 4F do not assay cell competition either since with this high recombination level, the number of hets will be small, and red cells left are not uniquely identified as heterozygotes. Figure 5 is more convincing, as it uses hsFlp to induce recombination.

2) It will be very useful to know if Myc induced super-competition is similarly affected by Nerfin. Myc overexpression presents its own issues, but perhaps inclusion of p35 will still trigger supercompetition (Levayer et al., Nature 2015). This can be combined with Nerfin RNAi. Alternatively, Yki overexpression could be combined with Nerfin RNAi.

3) Does Nerfin1 block transcription by blocking the interaction of Sd with DNA or by recruiting CtBP corepressors or both? Does Nerfin1 bind to DNA?

4) On a similar vein, to further strengthen the study, the authors need to demonstrate that the function of Nerfon-1 in cell competition is mediated by Sd.

5) The authors argue that Nerfin-1 and Tgi act in parallel through different mechanisms and also Tgi is not sufficient to explain the suppressor function of Sd, The one obvious experiment missing is the phenotype and the explanation for it, of the Nerfin-1 Tgi double mutant. This is easy to do, and the results from it need to be included in the manuscript.

---

## [Author Response]

[…] However and interestingly, the authors showed that Nerfin-1 plays a role in cell competition as Nerfiin-1 inactivation promotes the winner cells to eliminate loser cells. Finally, data are presented to show that the mammalian Nerfin-1 homolog INSM1 similarly binds to and inhibits TEAD dependent transcription.

The data presented are good quality and support the major conclusions in the paper. Although Vg/VGL4 have been shown to inhibit Sd/TEAD dependent transcription, the mechanism of Nerfin-1/INSM1 in transcription repression of Sd/TEAD is different. Therefore, the study by Pan and colleagues has revealed a new mechanism of Hippo pathway transcription regulation and has high potential significance.

Thank you for the positive comments on our manuscript. In this revision, we have conducted additional experiments in response to the reviewers’ comments. Specifically, we have added data showing 1) Nerfin-1 impairs the binding of Sd to the *diap1* HRE (Figure 3—figure supplement 2); 2) loss of *tgi* does not enhance the proliferation of *nerfin-1* mutant tissues, or the “super-winner” phenotype of *nerfin-1* mutant in cell competition (Figure 4—figure supplement 2F-H). We hope that these additional data/clarifications have significantly improved our manuscript.

Essential revisions:1) It is unclear that the cell lethality shown in Figure 4J-N is due to cell competition rather than a case of autonomous lethality of the homozygous cells. The efficiency of ey-flp is quite high and therefore, Figure 4E and 4F do not assay cell competition either since with this high recombination level, the number of hets will be small, and red cells left are not uniquely identified as heterozygotes. Figure 5 is more convincing, as it uses hsFlp to induce recombination.

The small number of heterozygous cells should not be a problem, since the eyeless-Flp system has been used extensively to study *Minute*-mediated cell competition (see for example, Tyler et al., 2007 and Lee et al., 2018, Dev Cell 46: 456-469). It is also worth noting that in such eyes, the red cells are unambiguously heterozygous for the cell-lethal mutation (*cl*) or *Minute*, since homozygous mutant cells of *cl* or *Minute* do not survive to adults. Therefore, this system allows one to unambiguously assess the relative fitness of WT cells versus *cl*/+ or *Minute*/+ cells. Irrespective of the baseline percentage of red cells in the eyeless-Flp system, the key is that we have uncovered increased occupancy of *nerfin-1* mutant cells compared to the WT cells, using both *cl* and *Minute* mutations (Figure 4). Importantly, similar results were obtained using hs-Flp-induced *Minute* wing discs (Figure 5E-F).

We acknowledge that unlike *Minute*, the interactions between WT and *cl*/+ cells are not classic examples of cell competition, although a recent paper from Hariharan lab has used this genetic setup to implicate Crumbs in cell competition (Hafezi et al., 2012). That is why we use *cl*/+ as an additional example to supplement our *Minute* experiment.

2) It will be very useful to know if Myc induced super-competition is similarly affected by Nerfin. Myc overexpression presents its own issues, but perhaps inclusion of p35 will still trigger supercompetition (Levayer et al., Nature 2015). This can be combined with Nerfin RNAi. Alternatively, Yki overexpression could be combined with Nerfin RNAi.

This paper focuses on the molecular characterization of Nerfin-1 as a novel Sd-Yki co-repressor, and we have already presented *in vivo* data showing the requirement of Nerfin-1 in two different models of cell competition. While we appreciate the reviewer’s suggestion to examine additional contexts of cell competition, we feel that these studies are beyond the scope of the current study. Along this line, we note that numerous genetic perturbations can all lead to cell competition, and currently it is unclear whether they all employ similar molecular mechanisms – for example, even for many of the better studied cell competition genes such as Xrp1, Sas and PTP10D, it is unclear whether they are involved in all contexts of cell competition.

3) Does Nerfin1 block transcription by blocking the interaction of Sd with DNA or by recruiting CtBP corepressors or both? Does Nerfin1 bind to DNA?

To address the reviewer’s questions, we performed ChIP analysis in S2R+ cells to study the binding of Sd and Nerfin-1 to the HRE site of the endogenous *diap1* locus (a known Sd-Yki target gene). We found that both proteins bind to the HRE site, and interestingly, the binding of Sd to HRE was decreased by the co-expression of Nerfin-1 but not a Nerfin-1 mutant protein defective in Sd-binding (Nerfin-1ZF1CA). These data are presented in a new figure (Figure 3—figure supplement 2) in our revised paper. Together with our data showing Nerfin-1-CtBP interactions, these results suggest that Nerfin-1 represses Sd-dependent transcription both by compromising Sd-DNA interaction and by recruiting CtBP corepressors.

4) On a similar vein, to further strengthen the study, the authors need to demonstrate that the function of Nerfon-1 in cell competition is mediated by Sd.

We appreciate the reviewers’ suggestion. Since Nerfin-1 functions in the winner cells only, in order to demonstrate that Nerfin-1’s function in cell competition is mediated by Sd, one needs to remove *sd* specifically in the winner cells. This experiment is technically challenging since *sd* is on the X chromosome and *nerfin-1* is on the 3^rd^ chromosome, and would require an *sd*-rescue transgene on the 3^rd^ chromosome, which are not available. As an alternative, we tried to do this by RNAi knockdown of *sd*. Unfortunately, we had difficulties recovering compound mutant flies containing UAS-sdRNAi and *nerfin-1/Minute* with multiple attempts. Despite that, given our biochemical characterization of Nerfin-1-Sd protein interaction, the observation that the Nerfin-1 overexpression phenotype is dependent on Sd, the observation that the *nerfin-1* cell competition phenotype is sensitive to Yki dosage as well as the elevated expression of Sd target *diap1* in *nerfin-1* mutant winner cells, we feel that there is reasonable support for Nerfin-1 function through Sd.

5) The authors argue that Nerfin-1 and Tgi act in parallel through different mechanisms and also Tgi is not sufficient to explain the suppressor function of Sd, The one obvious experiment missing is the phenotype and the explanation for it, of the Nerfin-1 Tgi double mutant. This is easy to do, and the results from it need to be included in the manuscript.

As suggested by the reviewers, we have generated the *nerfin-1 tgi* double mutant. We found that the *nerfin-1 tgi* double mutant tissues do not over-proliferate compared to *nerfin-1* mutant tissues. We also analyzed the *nerfin-1 tgi* double mutant in *Minute* and *cl*/+ mediated cell competition and found that loss of *tgi* does not result in a “super-winner” phenotype, nor does it enhance the “super-winner” phenotype of *nerfin-1*. These new data are included in Figure 4—figure supplement 2F-H in the revised paper. Taken together, these results suggest that Nerfin-1 and Tgi may be required in different contexts to suppress Sd function.